# Stomatal Traits and Barley (*Hordeum vulgare* L.) Forage Yield in Drought Conditions of Northeastern Mexico

**DOI:** 10.3390/plants10071318

**Published:** 2021-06-28

**Authors:** Foroughbakhch Pournavab Rahim, Torres Tapia María Alejandra, Zamora Villa Víctor Manuel, Treviño Ramirez José Elías, Ngangyo Heya Maginot

**Affiliations:** 1Biological Science School, Universidad Autónoma de Nuevo León, Ave. Pedro de Alba s/n cruz con Ave. Manuel L. Barragán, San Nicolás de los Garza 66451, Nuevo León, Mexico; rahim.forough@gmail.com (F.P.R.); atorres_tapia@hotmail.com (T.T.M.A.); 2Breeding Division, Universidad Autónoma Agraria Antonio Narro, Calzada Antonio Narro No. 1923, Col. Buenavista, Saltillo 25315, Coahuila, Mexico; zamora2602@yahoo.com.mx; 3Agronomy School, Universidad Autónoma de Nuevo León, Francisco Villa s/n, Col. Ex-Hacienda “El Canadá”, Escobedo 66050, Nuevo León, Mexico

**Keywords:** awnless barley, forage yield, stomatal index, chlorophyll content, NVDI, path coefficient

## Abstract

Infrared technology is a practical, fast, non-destructive method that helps in forecasting plant development and can be used to select physiological traits, instead of other methodologies that require more time and breeding efforts. According to the statistical analyses and the relationship between the direct and indirect effects of the variables, this technology could serve as the basis for implementing a genotype selection methodology. Awnless barley was assessed in a randomized block design with three replicates in two crop seasons at Mexico’s northeastern region. Two samplings were carried out during crop development: at 75 and 90 days after sowing. The infrared and stomatal sensors were used to identify the direct and indirect effects of stomata’s traits on dry forage yield. The data were analyzed in a subdivided plot design, using mean comparison tests, correlation coefficients and path analyses, finding significant differences (*p* < 0.05) among localities and among samplings. Dry forage yield was significant and positively correlated with plant height (r = 0.578) and canopy temperature (r = 0.724), and negatively correlated with the leaf upper side stomatal width (r = −0.409) and the leaf lower side stomatal width (r = −0.641), chlorophyll content and vegetation index. Temperature, chlorophyll, density and leaf lower side stomatal index had the strongest direct effects on yield. Therefore, the infrared technology appears as a way to select high yielding awnless forage barley, to obtain the correlation, the positive direct effect of temperature and the negative effect of chlorophyll. Due to their direct effects, low density and low leaf underside stomatal indexes can also help in the indirect selection of higher yielding forage barley genotypes.

## 1. Introduction

Drought is among the main abiotic stress that affects crop growth and productivity, leading to lower income for farmers. Climate change is expected to increase the global frequency and severity of drought events [1], causing high temperatures and scarce rainfall, which will dramatically increase and prolong drought, meaning that compared to it affecting 1–3% of the land in the present day, it could affect 30% by 2090 [2]. During severe water stress periods, the leaves restrict their water loss as an important survival mechanism by closing their stomata. However, early stomatal closure decreases net photosynthesis by reducing the photosynthetic activity of PSII, the amounts of fixed C and the activity of key photosynthetic enzymes, resulting in a decrease in leaf area, leaf width and mean area per mesophyll cell and eventually losses in grain yield [3]. Even though the total stomatal pore area is 5% of the leaf surface, transpirational water loss across the stomatal pores contributes to 70% of total water use by plants [4]. Therefore, an important aspect for increasing drought tolerance lies in a better understanding of the molecular mechanisms and genetic control of stomatal distribution and opening associated with growth rate and grain yield under abiotic stress [5,6]. Moreover, the implementation of crop management practices can potentially alleviate the harmful effects of drought and heat stresses, including soil management and culture practices, irrigation, crop residues and mulching, and selection of more appropriate crop varieties [7].

Barley is the fourth most important cereal crop in the world after wheat, maize, and rice, with up to 85% of the harvested barley used for animal feeding, including cattle, swine, and poultry [8,9], and it can be used as green fodder, hay, or can be harvested at maturity and threshed for grain and straw purposes, and sometimes its stubbles are grazed [10].

For the development and release of new barley forage genotypes with major drought tolerance traits, it is necessary to know the relationship between dry forage yield and its components, as well as the relationship between other determining variables, to facilitate the work of breeders in the development of the selection criteria. Some of the criteria consist of selecting awnless genotypes to avoid injuring the snouts of animals [11,12], or selecting superior higher yielding dry forage genotypes [13,14]. Nevertheless, it is not always easy to determine all the yield-related variables.

Usually, the dry matter content or aerial biomass is determined through destructive methods, harvesting and measuring the quantity of dry matter or dry forage at a given time. Recently, infrared spectral reflectance technologies have been used, such as the normalized difference vegetation index (NDVI), in order to measure crop growth, crop development and crop yield under simulated conditions [15]. The use of these spectral reflectance indices is a practical breeding method that allows for selecting specific physiological traits. For instance, it is possible to increase the yield in different crops by increasing the photosynthesis’ rate, because the production of dry matter fully depends on this process [16], and there is a genetic association between the chlorophyll content and yield [17,18]. There are some reports referring to the use of NDVI and the chlorophyll content to form groups of forage genotypes [14].

On the other hand, the path coefficient is a useful method to determine cause–effect relationships, which consists of conducting a statistical cause–effect analysis of correlated variables, analyzing the variables’ interdependency, as a supplement to regression and correlation studies [19]. In crop plant breeding, the study of interrelations among traits that determine yield has been done using correlation and path coefficients [20], in order to break down the direct and indirect effects of the correlations as a way to estimate the relative importance of causal factors. In grain barley, the direct effect of several agronomic variables on grain yield has been studied [21,22,23,24]. However, little is known about the direct effects of agronomic variables on forage and even less is known about the relationship between stomatal variables and forage yield. Therefore, the purpose of this research work was to study the relationships among stomatal traits and the variables determined through infrared sensors, in order to identify, according to the path analysis, the most relevant variables indicating direct and indirect effects on dry forage yield of awnless barley in drought conditions.

## 2. Results

### 2.1. Barley Stomatal Traits in Two Different Drought Conditions

The results show significant differences (*p* = 0.02) among localities for the stomatal density and index, and no significant differences (*p* = 0.09) within the sides of leaves, nor between samplings. Stomatal density registered values from 44.97 to 49.91 stomata/mm^2^ in locality 1 and from 66.10 to 96.37 stomata/mm^2^ in locality 2, with the highest value presented by the upper side foliar surface at 90 days after sowing, and the lowest value by the same upper side foliar surface, but at 75 days after sowing (Figure 1a). Regarding the stomatal index, the highest values were registered in locality 1 with the range from 77.79 to 81.27% and the lowest values in locality 2 with the range from 31.69 to 33.11% (Figure 1b).

No significant differences were found during the samplings (*p* = 0.07) in the stomatal dimensions among the localities or in the foliar surface. Stomata length ranged from 51.13 to 55.05 µm in locality 1, and from 49.59 to 51.57 µm in locality 2, while the stomata width ranged from 23.14 to 24.49 µm in locality 1, and from 19.32 to 24.54 µm in locality 2 (Figure 2). However, the width of the stomata on the upper side foliar surface was greater at 75 days after sowing (24.54 µm) than at 90 days after sowing (20.52 µm).

### 2.2. Barley Agronomic Variables in Two Different Drought Conditions

There were significant differences (*p* = 0.03) in the agronomic variables studied in this work, except for height, where there was no significant difference (*p* = 0.07). The canopy temperature was 17.22 °C at 75 DAS and 23.6 °C at 90 DAS in the locality 1, reaching up to 30 °C at 90 DAS in the locality 2 (Figure 3a). The NDVI index was higher in locality 1 than in locality 2, with values of 0.83 and 0.72 at 75 and 90 days after sowing, respectively, for locality 1, and values of 0.76 and 0.44 at 75 and 90 days after sowing, respectively, for locality 2 (Figure 3b). For the chlorophyll content index (CCI), locality 1 registered the same value (77.2 Spad) both at 75 and 90 days after sowing, while locality 2 showed a CCI of 38.75 Spad at 75 DAS, and 46.33 Spad at 90 DAS (Figure 3c). Regarding height, the values were similar between locality 1 and locality 2, higher at 90 DAS (113.33 and 111.67 cm, respectively) than at 75 DAS (97.35 and 96.25 cm, respectively), as indicated in Figure 3d.

In the case of yield, significant differences were recorded among localities and between sampling time (*p* = 0.01). Barley yielded more in locality 1 at 75 days after sowing (17.36 t. ha^−1^) than in locality 2, where only 6.38 t. ha^−1^ was recorded (Figure 4). At 90 days after sowing, the barley produced 9.56 t. ha^−1^ in locality 1, and 12.02 t. ha^−1^ in locality 2.

### 2.3. Relationship of Barley Stomatal Traits and Agronomic Characteristics in Drought Conditions

The correlation coefficients of Table 1 show that barley’s forage yield was associated in a significant and positive way with height (r = 0.578) and canopy temperature (r = 0.724), and in a negative way with the chlorophyll content index (r = −0.516), NDVI (r = −0.543) as well as the stomata width of the foliar upper side (r = −0.409) and lower side (r = −0.641), suggesting that the wider stomata on both surfaces of barley leaves are associated with lower dry forage yields. However, the other stomatal characteristics such as density, index and length of both the upper surface and the lower side of the leaf, did not have a significant association with barley forage yield.

On the other hand, canopy temperature showed a significant and positive relationship with height (r = 0.655), and a negative relationship with NDVI (r = −0.647), while the stomatal index of foliar lower side was significantly and negatively associated with height (r = −0.403).

Table 2 shows the direct and indirect effects of the variables obtained by means of the path analysis and the correlation coefficients with dry forage yield. The direct effects are shown in black on the main diagonal line. The higher direct effects on yield were obtained with canopy temperature (0.517), chlorophyll content index-CCI (−0.363), stomatal density of foliar lower side-SDL (−0.429) and stomatal index of foliar lower side-SIL (−0.331). The direct effect of height over yield was close to zero (0.025).

The values outside the main diagonal line are considered the indirect effects attributed to the interrelations among all the other variables and the last column shows the yield correlation coefficients and the variables, highlighting in bold letters those variables that were statistically significant: canopy temperature, NDVI, height, CCI, stomatal width of foliar upper side and stomatal width of foliar lower side.

## 3. Discussion

The significant differences registered between the localities could be associated with the particular conditions of each locality, which may have affected the expression of the stomatal variables, as has been mentioned by some authors [25,26]. However, the length and width of the stomata on the upper side of leaves did not show significant changes at different sampling times, and the same occurred with the stomatal density and the stomatal index, as well as the stomatal length of the lower side of barley leaves. These findings coincide with Wilkinson [27], who reported that these results are not distortions, but part of the plant system troubleshooting trait, which was confirmed by Quintana et al. [28] in *Coffea canephora* plants under different environments. The small changes that may occur at the stomata openings in terms of length and width of the protective cells and the stomata [29] can be considered variations of the stomatal size in response to the environmental conditions of each locality and the available water supplied during plant development. These changes can also be linked to changes in the photoperiod, water availability and soil conditions, and they play an important role in crop conditioning by maintaining or decreasing the stomatal density and the stomatal index in order to provide stomatal strength and avoid excess of transpiration [26].

The correlation coefficients showed a significant and positive relationship between barley’s yield and height, as also reported by Torres et al. [14] in forage barleys and in forage wheat by Zamora et al. [13]; and it has even been related to higher wheat grain yields, according to Meles et al. [30]; in oats and sorghum according to Gupta and Mahte [31] and Bhusal et al. [32]. On the other hand, the relationship between barley’s forage yield and canopy temperature, as well as its negative association with NDVI and with CCI, coincide with Lopes and Reynolds’ [33] reports on wheat production. Additionally, the coefficients between yield and the stomatal variables showed negative significance with the stomatal width, both on the upper side of the leaves as well as on the lower side, suggesting that the wider stomata on both surfaces of the tested barley leaves were associated with lower dry forage yields. Based on these results, it seems logical that the traits of narrower stomata on the leaves can be used as an indicator to determine dry forage yield. According to Limin and Fowler [34,35], there is a precedent in wheat where longer stomata were related to greater tolerance to freezing.

Another relationship was the significant and negative association of the stomatal density with the stomatal index of both the upper side and the lower side, similarly to what was found in barleys that were genetically manipulated to increase drought stress tolerance [36].

The direct effects of canopy temperature, chlorophyll content index, stomatal density of the lower side and stomatal index of the lower side on barley’s forage yield contrast with previous reports on forage oats and the direct effect of the plant’s height on forage yield [37]. This effect was also reported for sorghum and corn forage [32,38], different to what was found in the present research work, since the direct effect of height on yield was close to zero (0.025).

Since the stomatal widths of both the upper side and lower side have significant and negative correlation coefficients (r = −0.409 and r = −0.641), one may think that they also have high-value direct effects, as has been stated by some authors [32,37,38,39]. However, such values were low in this research work. Analyzing the components of these correlations, it appeared that the indirect effects of canopy temperature (−0.134) and CCI (−0.204) on stomatal width of the upper side were the greatest contributors to the expression of the detected correlation coefficient, similarly to the indirect effects of canopy temperature (−0.219) and CCI (−0.214) over stomatal width of the foliar lower side. This suggests that lower temperature of the canopy and CCI have some impact on the stomatal width, in such a way that plants with lower canopy temperatures developed a more efficient cooling system with narrower stomata, as found in other stomatal research works performed in mutant barleys [36] and in *Coffea canephora* [28]. Furthermore, these results suggest that there is a certain effect of the chlorophyll content (CCI) on the stomatal width in both leaf surfaces, which is a response that has been detected before in other plant species [26,28,40].

Additionally, there were higher, direct and positive effects of canopy temperature (0.517) on dry forage yield, and it is the main contributor to the correlation detected between both variables. In the same way, canopy temperature had an important indirect effect on the NDVI (−0.334), height (0.338), stomatal index of the lower side (−0.128), and the stomatal width of the upper (−0.134) and lower side (−0.219), which are significantly associated with forage yield, coinciding with the temperature change associated with the stomatal width, as reported by Pandey et al. [29]. These results allow us to confirm the sensitivity of the infrared technology to different climate conditions reflected at the level of the canopy’s temperature, as mentioned by Doraiswamy et al. [15]. 

The second highest direct effect (although negative) on yield was the stomatal density of foliar lower side (−0.429), but it did not show any significant association with the forage yield. This result differs from the reports of Hughes et al. [36], who mentioned that when the stomatal density decreases, the drought stress tolerance increases, without a reduction in barley’s biomass. However, their work was conducted under greenhouse conditions and their determinations were made in younger juvenile stages. Due to the stomatal density of the lower side’s (SDL) direct effect on forage yield, lower SDL can produce higher forage yields, probably because the plants use water more efficiently, as has been documented in wheat [41]. At the level of correlation, no significant association of SDL with forage yield was found, maybe due to the positive indirect effect of the stomatal index in foliar lower side (0.268). 

The chlorophyll content index (CCI) presented the third direct effect over dry forage yield, with a negative significant correlation coefficient (r = −0.516), which coincides with the affirmation that foliar chlorophyll concentration can be used to indirectly estimate the biomass content [42].

The fourth direct effect came from the foliar lower side stomatal index (SIL) with a value of −0.331, presenting a correlation coefficient without any significance in terms of yield. This result is similar to the response produced by the foliar lower side stomatal density, where SDL became the cause with an indirect positive effect. These indirect effects between SDL and SIL are explained to a great extent by the method in which the index is calculated, since stomatal density is involved in both the dividend and in the divider of the formula that was used. In that regard, Barrientos-Priego et al. [43] mentioned that the stomatal index is a good indicator that can be used to differentiate avocado seedlings for selection purposes, because this index had little variation at different leaf positions on the stem, while stomatal density had more variation. 

## 4. Materials and Methods

### 4.1. Study Area

The experimental work was performed during the fall–winter 2018–2019 crop season in two localities: (1) Rancho Beta Santa Mónica, located at Ejido San Lorenzo, in the Municipality of San Pedro de las Colonias, Coahuila de Zaragoza, Mexico, at coordinates 25°43′26.0″ north latitude and 103°09′12.0″ west longitude. The site had a mean altitude of 1090 m above sea level, a mean annual temperature of 21.2 °C, and a mean annual rainfall of 181 mm. (2) Experimental field at Zaragoza, Coahuila, located at 28°30′ north latitude and 100°55′ west longitude, at a mean altitude of 360 m above sea level. Mean annual temperature of 20 °C and mean annual rainfall of 300 to 400 mm. 

### 4.2. Sampling Preparation and Experimental Design

The samples consisted of awnless forage barley (awnless spikes) resulting from crossing GABYAN95 cultivar (released by “Universidad Autónoma Agraria Antonio Narro”) with “Esperanza” commercial cultivar, released by “Instituto de Investigaciones Forestales, Agrícolas y Pecuarias” (Forestry, Agricultural and Livestock Research Institute). 

The land was prepared following the traditional cultural practices for establishing small winter grain cereals in regions where cropping is done under irrigation, including fallow, disc harrowing, grading and furrow clearing. Planting was done by hand at a density of 83.33 kg ha^−1^ by depositing the seeds evenly at the bottom of the furrow. There were 3 plots with 6 furrows of 3.0 m in length and a space of 0.36 m between furrows in each locality (drought condition). 

The experiment was carried out in a fully randomized block design with three replicates in each locality, according to the procedure described by Zar [44]. Since randomized block distribution is a quite efficient and commonly used method, more benefits can be obtained through the variation gradient, by forming blocks distributed perpendicularly to the gradients’ direction. 

Fertilization was split into 120 Kg N ha^−1^, 60 kg P ha^−1^ and 0 Kg K ha^−1^, supplying half the nitrogen and all the phosphorus at planting and the rest of the nitrogen during the first auxiliary irrigation, using urea and MAP (NH_4_H_2_PO_4_) as nitrogen sources, the latter being the source of phosphorus. Irrigation was applied at planting and on two other occasions as auxiliary irrigation before sampling, according to the crop status throughout the development. 

Two forage sampling procedures were conducted, the first one at 75 days after planting and the second at 90 days after planting, once the flag leaf had fully emerged. In this process, 0.5 linear meters of a furrow at full competition were sickled by hand at 5 cm above the ground. The harvested forage was kept in kraft paper bags before drying and weighing them (t ha^−1^).

### 4.3. Evaluation of the Barley Stomatal Traits

In order to measure density and the stomatal index of the upper side and the lower side of leaves, three flag leaves were cut from each plot in every sampling, and the epidermal impressions were taken from the upper side surface (adaxial) and the lower side surface (abaxial) by applying semi-liquid xylol-polystyrene on the foliar surface with a brush. The film was removed after drying with clear adhesive tape and it was placed on a slide. Every impression was observed at random in three microscopic fields at 40X, using a compound Carl Zeiss microscope, assessing nine fields on the foliar upper side surface and another nine fields at the lower side of leaves per plot. A micro-photograph was taken from every field using a PixeraWinder Pro digital camera to determine the stomatal density of the upper side and the lower side of leaves (SDU and SDL) by counting the stomata number per observed field, in the following way: SD = stomata number/0.02479 mm^2^ (surface area of the photograph) = stomata per mm^2^ [45]. The foliar upper side and lower side stomatal indexes (SIU and SIL) were measured according to Wilkinson [27] using Equation (1).
(1)$$\mathrm{SI}=\left(\frac{NS}{NEC}+NS\right)\times 100$$
where *NS* is the number of stomata on the surface area per field of observation and *NEC* is the number of typical epidermal cells on the surface area at the field of observation.

A measuring software AxionVision Rel. 4.8, Digypro4 was used to measure the length and width of the occlusive cells, including the pore that forms part of the stomatal apparatus, the stomata on the upper side of the leaves’ surface (SLU and SWU) and the stomata on the lower side (SLL and SWL), according to Ramírez et al. [25].

### 4.4. Determination of the Barley Forage Production and Quality

The following variables were recorded at sickling time. Plant height (H) was measured in centimeters from the ground surface to the upper part of the plant located in the middle of each experimental plot using a measuring tape. Canopy temperature (T°) was measured through an infrared IP-54 Fluke thermometer in the middle of each experimental plot, in degrees Celsius. The chlorophyll content index (CCI) was determined through a SPAD-502 Konica Minolta chlorophyll meter, measuring the chlorophyll content or the “green color” in plants at a scale of 0–0.99. Since this value is proportional to the quantity of chlorophyll in the sample, the reading was realized at the center of the flag leaf, assessing 10 leaves from each experimental plot. The normalized difference vegetation index (NDVI) was measured through a portable GreenSeeker sensor that releases short red and infrared light blasts to determine the level of reflectance at the center of each plot. Half a meter of plants was cut from the two central furrows of each plot, using a chaser 5 cm from the ground and placing it in a brown paper bag, each plot sample was weighed with a Torrey brand electronic scale, recording the fresh weight in grams. The sample was left to dry at room temperature under a laminated roof for a week, and weighed on the electronic scale, recording the dry weight of the plant and the yield in Kg. ha^−1^ was calculated.

### 4.5. Statistical Analysis

Data obtained from the field and laboratory were analyzed using a sub-divided plot design, considering localities as large plots, samplings as medium-size plots and the studied species as small plots. Means were compared using the minimum significant difference (DMS, α ≤ 0.05) and Student’s *t*-test was performed to determine differences among the two drought conditions (localities L1 and L2). Pearson’s correlation coefficients were determined from averages in localities and by the sampling time. Direct and indirect effects were determined based on the correlations, using the path analysis proposed by Wright [20], building a correlations matrix among the studied variables and a yield-correlation’s vector, in order to find a matrix-based solution with a routine developed in SAS 9.0 [46], through an IML procedure.

## 5. Conclusions

The infrared technology was use as a tool to determine awnless forage barley yield, that was directly related to plant canopy temperature, and negatively to chlorophyll content index (CCI). Regarding the stomatal study, due to their direct effect on yield, low stomatal density values and the stomatal index of the foliar lower side can also be considered as indicators for forage barley’s yield, as well as stomatal width and the normalized difference vegetation index (NDVI), allowing stomatal traits to act as key determinants of growth rate and water balance in plants.

## Figures and Tables

**Figure 1 plants-10-01318-f001:**
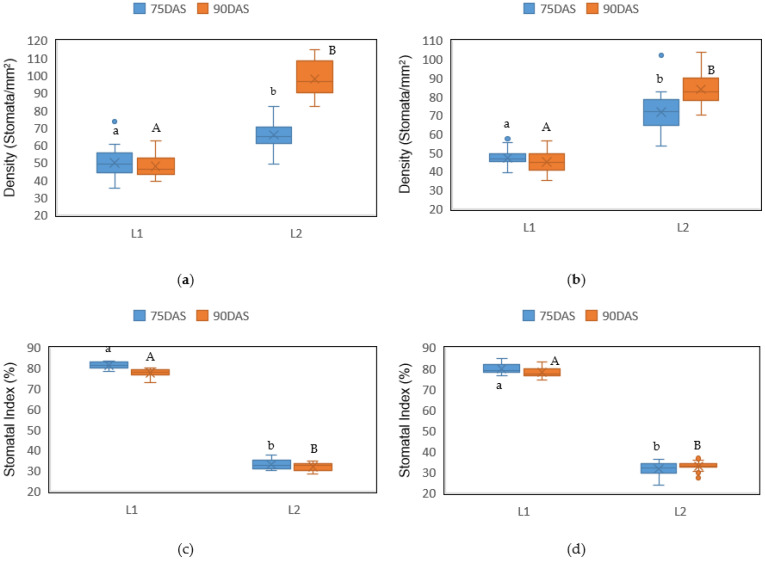
Stomatal density of the upper side (**a**) and lower side (**b**) and stomatal index of the upper side (**c**) and lower side (**d**) of barley leaves at 75 and 90 days after sowing (DAS) in two drought conditions (L1 and L2). The letters indicate the statistical differences provided by Student’s *t*-test, among the two drought conditions.

**Figure 2 plants-10-01318-f002:**
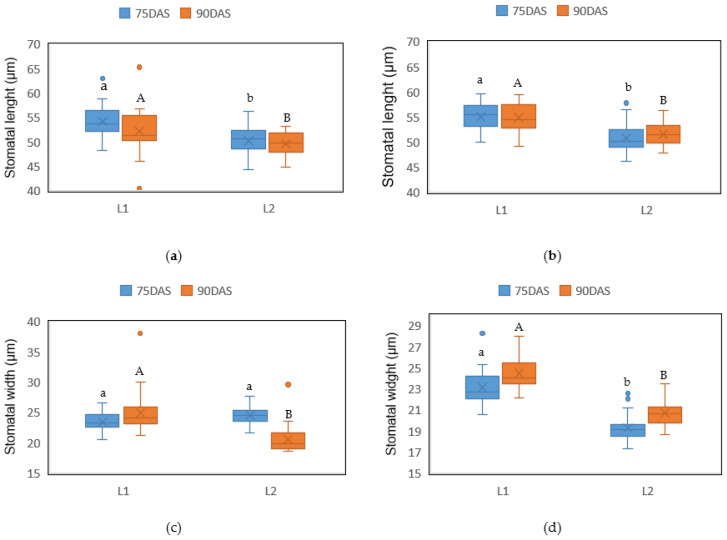
Stomata dimensions: (**a**) stomata length of the upper side, (**b**) stomata length of the lower side; (**c**) stomata width of the upper side, (**d**) stomata width of the upper side of barley leaves at 75 and 90 days after sowing (DAS) in two drought conditions (L1 and L2). The letters indicate the statistical differences provided by Student’s *t*-test among the two drought conditions.

**Figure 3 plants-10-01318-f003:**
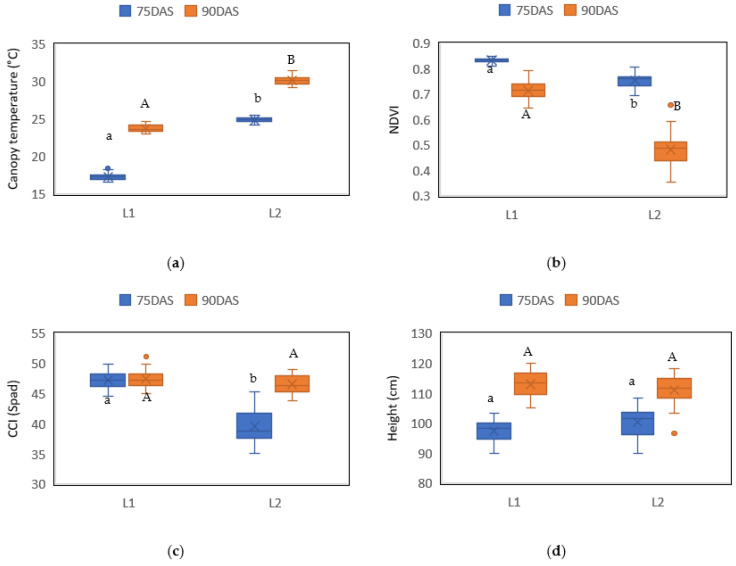
Agronomic variables of barley in two drought conditions (L1 and L2) at 75 and 90 days after sowing (DAS). (**a**) Canopy temperature; (**b**) NDVI; (**c**) chlorophyll content index (CCI) and (**d**) height. The letters indicate the statistical differences provided by Student’s t-test among the two drought conditions.

**Figure 4 plants-10-01318-f004:**
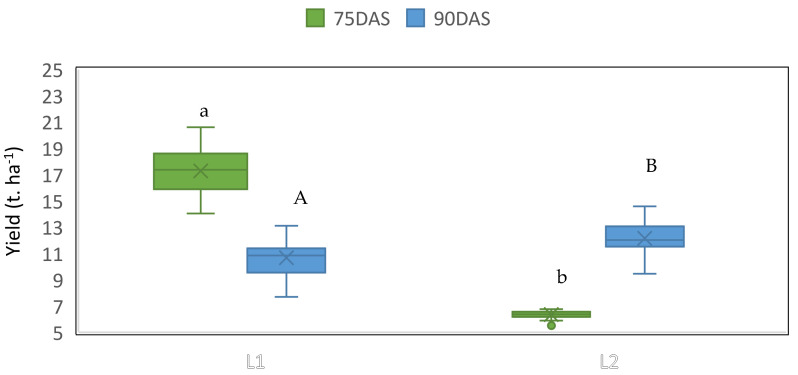
Barley yield at 75 and 90 days after sowing (DAS) in two different drought conditions (L1 and L2). The letters indicate the statistical differences provided by Student’s *t*-test, among the two drought conditions.

**Table 1 plants-10-01318-t001:** Correlation coefficients among barley stomatal traits and agronomic variables in drought conditions.

	T°	NDVI	H	CCI	SDU	SIU	SLU	SWU	SDL	SIL	SLL	SWL	Yield
**T°**	1.000	**−0.647**	**0.655**	−0.083	0.197	−0.144	−0.085	−0.260	0.141	−0.247	−0.183	**−** **0.423**	**0.724**
**NDVI**		1.000	−0.380	−0.053	−0.014	−0.274	−0.304	0.156	0.154	−0.097	−0.086	0.174	**−** **0.543**
**H**			1.000	−0.160	0.105	−0.236	0.004	0.044	0.171	**−** **0.403**	−0.076	−0.290	**0.578**
**CCI**				1.000	**−** **0.435**	**0.479**	0.112	**0.563**	0.185	0.044	−0.001	**0.589**	**−** **0.516**
**SDU**					1.000	**−** **0.653**	**−** **0.515**	−0.342	0.255	−0.277	−0.297	**−0.537**	0.272
**SIU**						1.000	**0.415**	0.204	−0.084	0.116	0.019	0.229	−0.184
**SLU**							1.000	0.302	−0.090	0.106	**0.535**	**0.498**	−0.021
**SWU**								1.000	0.073	−0.038	0.306	**0.719**	**−** **0.409**
**SDL**									1.000	**−** **0.812**	−0.244	0.024	−0.144
**SIL**										1.000	0.266	0.239	−0.175
**SLL**											1.000	**0.582**	−0.185
**SWL**												1.000	**−** **0.641**
**Yield**													1.000

The values in bold letters are statistically significant at 0.05 % probability. T° = Temperature of the canopy; NDVI = Normalized Difference Vegetation index; H = plant height; CCI = chlorophyll content index; SDU = stomatal density of foliar upper side; SIU = stomatal index of foliar upper side; SLU = stomatal length of foliar upper side; SWU = stomatal width of foliar upper side; SDL = stomatal density of foliar lower side; SIL = stomatal index of foliar lower side; SLL = stomatal length of foliar lower side; SWL = stomatal width of foliar lower side; Yield = dry forage yield.

**Table 2 plants-10-01318-t002:** Direct and indirect effects of the barley’s test variables over dry forage yield.

	T°	NDVI	H	CCI	SDU	SIU	SLU	SWU	SDL	SIL	SLL	SWL	r
**T°**	**0.517 ***	0.094	0.016	0.030	0.004	−0.001	−0.011	−0.012	−0.060	0.082	0.026	0.040	**0.724 ***
**NDVI**	−0.334	**−0.146 ***	−0.009	0.019	0.000	−0.001	−0.041	0.007	−0.066	0.032	0.012	−0.016	**−0.543 ***
**H**	0.338	0.055	**0.025 ***	0.058	0.002	−0.001	0.001	0.002	−0.073	0.133	0.011	0.027	**0.578 ***
**CCI**	−0.043	0.008	−0.004	**−0.363 ***	−0.009	0.003	0.015	0.026	−0.079	−0.015	0.000	−0.055	**−0.516 ***
**SDU**	0.102	0.002	0.003	0.158	**0.021 ***	−0.003	−0.069	−0.016	−0.109	0.092	0.042	0.050	0.272
**SIU**	−0.074	0.040	−0.006	−0.174	−0.014	**0.005 ***	0.055	0.010	0.036	−0.038	−0.003	−0.022	−0.184
**SLU**	−0.044	0.044	0.000	−0.041	−0.011	0.002	**0.133 ***	0.014	0.039	−0.035	−0.077	−0.047	−0.021
**SWU**	−0.134	−0.023	0.001	−0.204	−0.007	0.001	0.040	**0.047 ***	−0.031	0.013	−0.044	−0.068	**−0.409 ***
**SDL**	0.073	−0.022	0.004	−0.067	0.005	0.000	−0.012	0.003	**−0.429 ***	0.268	0.035	−0.002	−0.144
**SIL**	−0.128	0.014	−0.010	−0.016	−0.006	0.001	0.014	−0.002	0.348	**−0.331 ***	−0.038	−0.022	−0.175
**SLL**	−0.095	0.013	−0.002	0.000	−0.006	0.000	0.071	0.014	0.105	−0.088	**−0.143 ***	−0.055	−0.185
**SWL**	−0.219	−0.025	−0.007	−0.214	−0.011	0.001	0.066	0.034	−0.010	−0.079	−0.083	**−0.094 ***	**−0.641 ***

r = Correlation coefficient (* in bold are for variables that were statistically significant at 0.05). T° = Temperature of the canopy; NDVI = Normalized Difference Vegetation index; H = Plant height; CCI = Chlorophyll content index; SDU = stomatal density of foliar upper side; SIU = stomatal index of foliar upper side; SLU = stomatal length of foliar upper side; SWU = stomatal width of foliar upper side; SDL = stomatal density of foliar lower side; SIL = stomatal index of foliar lower side; SLL = stomatal length of foliar lower side; SWL = stomatal width of foliar lower side; Yield = dry forage yield.

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
