# Peer review of "Stomatal Traits and Barley (Hordeum vulgare L.) Forage Yield in Drought Conditions of Northeastern Mexico"

_plants, 2021, doi:10.3390/plants10071318_

Round 1

Reviewer 1 Report

Review comments:

This article entitled “Stomatal Traits and Barley (Hordeum Vulgare L.) Forage Yield in Drought Conditions of Northeastern Mexico” fits the scope of this journal. But I will still propose two general comments (1) short and reorganize the long sentence; (2) polish the language.

L17: “can be used to selecting” = “can be used to select”

L18-L20: bad grammar. “link” and “breakdown” should be of the same voice, while “basis” should be “the basis”

L23-L26: break this long sentence down to 2 sentences.

L41-42: “the leaves restrict their water loss as an important survival mechanism, closing their stomata.” The authors used similar expressions in multiple places. I personal do not like this, one alternative should be “by closing their stomata”.

L72: “indexes” should be “indices”

L78: First “in the other hand” should be “on the other hand”, second even for “on the other hand” I will not suggest to use such as phase in an academic journal. You can use “Additionally”.

L92: Section 2. We all know that p=0.05 is the threshold for significant and non-significant differences. Thus, I will suggest the author to put the p value based on their results. For example, if based on data analysis, p=0.01<0.05, the all the audience will know there is a significant difference.

L137: Section 2.3. It is always good to study the correlation between variables, However, based on my experience, I do not think r is a good way to illustrate “significant differences”. I will recommend to compute p value instead, using F-test.

L203: “there were …. effect” should be “there were … effects”.

L264: Section 4: it is awful to put material and method section after the result section. I am not sure if this is the requirement of this journal, but when I review this manuscript. I have to read Section 4 first in order to understand Section 2

Author Response

Thanks to reviewer 1, for his valuable comments, which have helped significantly improve our manuscript.
His recommendations have been taken into account, as can be seen in the new version resubmitted to the journal.

Reviewer 2 Report

In the present mansucript, the authors study the relatonships among stomatal traits, density, size, etc.) and non-destructive infrared spectral reflectance in order to predict crop growth and productivity during drought stress in barley.

The introduction is well written, it references to previous studies. At the end of the introduction, together with the purpose of the research study, I would like to ask the authors to add an hypothesis. Please, describe your hypothesis and fundament it just in a few lines. Later, in the discussion, please, refer to your hypothesis. 

L84: Please, together with the citations 21-24, write also the 'agronomic variables' that these studies investigated.

Figures 1 to 4: I am not that happy when I see poorly worked graphics. I do not mean the the authors should use another software other than excel, openoffice, etc. But, at least, they could wrok better the design. I take Figure 1a as an example: 

  • y-axis, lower to the amount of labels to 5-7. not 30, 40, 50, 60... but 20, 40, 60, 80, 100, 120.
  • Change the scaling of graphs like Figure 3 to see better the boxplots.
  • Explain in the legend that this is a boxplot representing precentiles, x to x, with outliers if in the extreme XX % of the distribution. Say that the mean is represented as an "x", and the median as a horizontal bar,...
  • Remove the horizontal grey lines in the background at every label.
  • Make the axis lines black coloured instead of the default grey. Maybe try to close the graph as a rectangle with black lines in all sides.
  • Remove the grey line that frames the whole A and B panels of the figure. This copy-pasted format is not looking good.
  • A lot of space is not used in the graph because the boxplots are small in size (see Figure 2).
  • Explain in the legend that L1 and L2 are two drought conditions and which conditions they are. The figure must be stand-alone. Think that the reader did not arrive to the methods section yet.
  • Please, add statistical tests (Student's t-test, I suppoese) of differences among L1 and L2 with letters (a), (b) if the drought conditions had a difference, that it looks like yes. Don't forget to also mention in the legend of this figure and in the methods section.
  • Without these formatting changes, the manuscript should not be accepted. Otherwise it may seem to a reader that the study is of low quality or made by an undergraduated student. I apologize for the comparison. I just would like the manuscript to improve in its presentation.
  • I would have made four pannels in Figure 1 instead of two panels. SDU, SDL, SIU, SIL. But that's a choice of the authors. Please, consider doing the change.

I think that the authors should make additional  figures of the most important correlations, and not only showing in the tables. In this way, the reader can assess whether these correlations are linear or non-linear. And the strenght of them.

The reader needs to see with how many points the correlations are made. The discribution, etc.

Every correlation "r", must be followed by a P-value. This is such a basic practice that I can't understand why it is not yet here in the submitted manuscript. The reader must be able to know whether the correltion is significant or not.

Sentences like in L219 don't have any value if r = -0.134 and r=-0.204, apart from being low correlations, they are not significant.

Table 2: are the horizontal variables the direct or the indirect? Pleaes, specify in the legend.

Methods:

Section 4.4. is very narrow. We do not know how many measurements were made with the CCI, for example.

How many replicated measurements were averaged. one per leaf, then 10 measurements?

Was the barley harvested, dried, etc. to measure production? in an oven? for how long? when was the harvest done? at the end of the season? after flowering? This section is very poor.

This section deserves to be much lnger with different paragraphs for each variable.

Author Response

Thank you very much to Reviser 2 for his comments which are so relevant and important for the improvement of our manuscript. They have been taken into account, as can be seen in the new version.

Round 2

Reviewer 2 Report

I would appreciate a detailed point-per-point response to my review. Otherwise, I must make a big effort to discover myself whether the authors really took my comments in account and took action. The type of response that the authors made is not acceptable for a major revision.